# Think Big, Start Small: How Nanomedicine Could Alleviate the Burden of Rare CNS Diseases

**DOI:** 10.3390/ph14020109

**Published:** 2021-01-30

**Authors:** Abdelfattah Faouzi, Valérie Gaëlle Roullin

**Affiliations:** 1Center for Clinical Pharmacology, St. Louis College of Pharmacy and Washington University School of Medicine, St. Louis, MO 63131, USA; afaouzi@wustl.edu; 2Laboratoire de Nanotechnologies Pharmaceutiques, Faculté de Pharmacie, Université de Montréal, Montréal, QC H3T 1J4, Canada

**Keywords:** CNS disorders, rare pathologies, orphan diseases, nanotechnologies, nanomedicine, drug delivery systems

## Abstract

The complexity and organization of the central nervous system (CNS) is widely modulated by the presence of the blood–brain barrier (BBB) and the blood–cerebrospinal fluid barrier (BCSFB), which both act as biochemical, dynamic obstacles impeding any type of undesirable exogenous exchanges. The disruption of these barriers is usually associated with the development of neuropathologies which can be the consequence of genetic disorders, local antigenic invasions, or autoimmune diseases. These disorders can take the shape of rare CNS-related diseases (other than Alzheimer’s and Parkinson’s) which a exhibit relatively low or moderate prevalence and could be part of a potential line of treatments from current nanotargeted therapies. Indeed, one of the most promising therapeutical alternatives in that field comes from the development of nanotechnologies which can be divided between drug delivery systems and diagnostic tools. Unfortunately, the number of studies dedicated to treating these rare diseases using nanotherapeutics is limited, which is mostly due to a lack of interest from industrial pharmaceutical companies. In the present review, we will provide an overview of some of these rare CNS diseases, discuss the physiopathology of these disorders, shed light on how nanotherapies could be of interest as a credible line of treatment, and finally address the major issues which can hinder the development of efficient therapies in that area.

## 1. Introduction

Early in the 1990s, the development of nanomedicine arose as a very promising novel technique with several therapeutical benefits, and more specifically in cancer, neurodegenerative and infectious diseases [1,2,3]. This term includes a large range of “nano” objects which are characterized by a nanometric scale, such as nanoparticles (NPs), nanocarriers, or even nanodrugs [4,5]. The extremely small size of nanotherapies certainly make them promising candidates for the treatment of any disorder related to the central nervous system (CNS). Indeed, the blood–brain barrier (BBB) is a complex structure composed of specialized endothelial cells (ECs) tightly anastomosed by specific tight adherens junctions. The presence of these very tight, selective junctions is well known to restrict the passage of the majority of pathogens, as well as large and hydrophilic molecules, while allowing the entry of small non-polar and hydrophobic agents. This creates a huge obstacle when targeting CNS disorders, as more than 98% of neurotherapeutic drugs are excluded from the brain by the BBB [6]. Those ECs are surrounded and supported by pericytes and astrocytes, which form an additional physical layer. Furthermore, this multicellular architecture is reinforced by a series of physical and biological obstacles, such as the presence of elevated transelectrical resistance and an active efflux protein pump, contributing to further restricting the access of drugs to the CNS and an effective immunological defense based on microglia [7]. Therefore, the cellular complexities of the CNS and the spinal cord complicate even more the targeting of a specific cell population. As such, the use of nanotherapies appeared as a primary tool in order to overcome these issues. First of all, these agents can be designed in such a way that allows them to efficiently cross the BBB (Table 1). Due to their versatility, nanocarriers can display tridimensional shapes and suitable dimensions favoring translocation across the BBB [8,9,10,11]. Other nanoparticular features, such as the ability to change their surface properties—i.e., tuning the surface charge or decorating it with appropriate ligands/functionalized moieties (surfactants or polymers, for example)—may also allow valuable interactions at the luminal plasma membrane of endothelial cells, resulting in enhanced transcytosis across the BBB [12]. This can be the case of a nanocarrier which undergoes receptor-mediated transcytosis, which is achieved thanks to a genetically fused transporter protein or by promoting the passive diffusion of small lipophilic molecules through endothelial cells [13]. Another interesting feature comes from their ability to enhance or sustain drug release, which can be critical in order to insure proper drug bioavailability to the brain. Finally, nanotherapies can be engineered to target specific cells, tissues, biochemical pathways, or receptors. We listed in Table 1 the current nanocarriers developed for the treatment of CNS diseases.

Nanotherapies can therefore be applied to the treatment of rare diseases, especially for the CNS. A rare (or orphan) disease is usually caused by a very distinct genetic mutation and is defined by a prevalence affecting under 200,000 people in the United States, while it would be 1:2000 people in the European Union [14,15]. It is estimated that approximatively 30 million Americans, half of them children, currently suffer from a rare pathology [16]. A large group of rare diseases are neurological disorders which can affect the nervous system, including the brain, the spinal cord, and all the nerves running throughout the human body. These pathologies can be divided between two categories: the ones targeting the CNS (brain and spinal cord) and the ones targeting the peripheral nervous system. The National Institute of Health (NIH) database lists more than 1000 nervous system pathologies (exactly 1248, as retrieved in November 2020), among which the majority is considered to be rare diseases [17]. These neurological issues can originate from multiple causes, including genetic disorders, congenital abnormalities, infections, and lifestyle and environmental health problems (including malnutrition), or are caused by brain, spinal cord, or nerve injuries. In addition, they can display several different symptoms, such as paralysis, muscle weakness, poor coordination, loss of sensation, seizures, confusion, pain, and altered levels of consciousness, all of which are usually chronically debilitating and life-threatening for the patients. These symptoms, along with the scarce to non-existent effective curative strategies, complicate even more the treatment of these rare pathologies. Drug delivery methods, such as intravenous (iv.) injections, face other challenges, such as, for instance, the clinical confirmation of a treatment and the measurement of its efficacy. Indeed, it is often necessary to collect tissue samples in order to acknowledge diagnosis or disease progression or monitor the potency of a given therapy. These are generally not easily manageable when targeting the brain, except post-mortem. Even though Magnetic Resonance Imaging (MRI) and Positron Emission Tomography (PET) imaging have undergone tremendous improvement in performance in these past few years (especially for the diagnosis of Alzheimer’s and Parkinson’s diseases), some limitations still remain, such as radiation exposure, overall extremely high costs, and equipment accessibility [18].

Furthermore, the lack of relevant models for a specific disease, especially for understudied rare diseases, is often an issue and must be thoroughly addressed to enhance the in vivo predictive validity and (pre)clinical trial efficiency [19]. This could also help us to assess if the integrity of the BBB is maintained prior, during, and after administration of a nanotherapeutic tool, which is of tremendous importance when dealing with CNS disorders. Unfortunately, this feature is often absent from research data or improperly examined with dyes for instance [20]. A relevant illustration of this latter point would be the use of cationic nanoparticles which have long been demonstrated to disrupt the BBB, hence inducing neurotoxicity; consequently, the effect on the BBB integrity should be automatically assessed when dealing with this type of nanoformulation [21]. The same goes for certain classes of agents, such as psychostimulants or alcohol, which are known to alter the BBB integrity, resulting in neuroinflammation and subsequent toxicity [22].

Above all, drug development for CNS disorders is confronted with an additional major hurdle, regarding how to get therapies past the BBB. While this issue has been widely studied and described in the literature, even with very small molecules drugs are often blocked from entering the brain by the BBB. The use of nanovectors/nanocarriers has long been reported as a promising approach to help macromolecules reach the brain effectively; in that respect, Figure 1 presents a comparative evaluation of the overall curative benefits of nanomedicine versus conventional therapeutics (Figure 1) [23]. Nanoengineered materials can simultaneously sustain drug release, improve bioavailability, and safeguard active compounds from degradation. Unfortunately, only a small number of studies related to rare pathologies of the CNS and nanotherapeutics are presently described in the literature. To illustrate this point, a rapid PubMed survey covering the terms ((brain OR CNS) AND (rare disease) AND (treatment) AND (nanotherapy OR nanomedicine OR nanoparticles) AND NOT (brain tumor OR Parkinson’s OR Alzheimer’s OR Multiple Sclerosis)) only led to 25 publications, compared to 900 when applied to brain tumors or 519 with Parkinson’s disease (PubMed search motor, November 2020). Comparatively, nanotherapy/nanomedicine represents only 0.07% of published treatment options for CNS rare diseases, whereas it is 0.80% and 0.78% for brain tumors and Parkinson’s, respectively. The substantial impact of big pharmaceutical companies and their lack of interest in these diseases play a major part in that respect [24]. Indeed, the development of such technologies requires high costs for R&D and subsequent production which would not be economically beneficial, owing to the relatively small number of prospective patients.

However, many incentives originating directly from the Food and Drug Administration (FDA) or the European Medicine Agency (EMA) have been launched in recent years, which will hopefully help in stimulating this field and encourage efforts from the entire scientific community [25]. A great example of nanotechnology oriented toward the treatment of a rare disease (excluding the CNS field), would be the Lysodase agent, designed as a specific treatment for Gaucher’s disease in the early 2000s [26,27]. This inherited pathology results from a genetic deficiency of the glucocerebrosidase enzyme, which leads to glucocerebroside abnormal accumulation and ultimately hepatosplenomegaly, along with low blood and platelet counts. Gaucher’s disease can be classified into three different types: (1) non-neuronopathic, the most common form which does not affect the CNS and can develop at any age; (2) the most severe acute neuronopathic form, which is usually lethal and appears prenatally; (3) the subacute neuronopathic type. There is currently no curative strategy for Gaucher’s disease, with most of the treatments being mainly supportive, such as, for instance, bone marrow transplant aimed at correcting glucocerebrosidase deficiency. Enzyme replacement therapies (ERTs), or, more recently, oral substrate reduction therapies, are other strategies approved by the FDA with the poly(ethylene glycol) (PEG)-Glucocerebrosidase/Lysodase drug being under clinical investigation and showing significant promise, with a reduction in skeletal complications [27,28,29]. However, the costs of such alternatives often exceed hundreds of thousands of USD per year when administered iv. The use of nanotechnology can provide, in this case, specific nanocarriers which will allow efficient transport through the BBB without alteration or loss of activity and with sustained delivery [30].

Although there are currently several articles and reviews highlighting the role and scope of nanotherapies in modern medicine, to our knowledge none focus solely on the impact of nanomedicine and rare diseases of the CNS. Therefore, the primary aim of this article is to provide a non-exhaustive list of CNS-related rare diseases for which nanotherapies could display potential avenues and describe the limitations and benefits that nanotechnologies could offer in order to treat these pathologies. Finally, we will also give an insight into their significance in treating these neurological disorders and discuss opportunities for future research.

## 2. Nanocarriers Used for the Treatment of CNS Disorders and Strategies to Bypass the BBB

One of the major obstacles in the development of effective CNS treatments is the presence of the tightly selective BBB. As such, the use of different nanocarriers (listed in Table 1) along with different administration routes can be of great interest in order to bypass the BBB.

### 2.1. Parenteral Drug Delivery

As for many pathologies scattered elsewhere in the human body, the preferred route of administration for drugs intended for CNS pathologies is intravenous administration (iv.), which is minimally invasive and allows complete control over bioavailability, a precise dosage of often-toxic drugs, and repeated administrations of several-hour infusions, achieving a near-constant plasma concentration carefully located in the therapeutic range. However, and in contrast with other organs or tissues, drugs administered intravenously should present either an increased ability to passively diffuse throughout the BBB (such as, for instance, carmustine and temozolomide, two very hydrophobic drugs) or use one of the assisted transcytosis pathways, as schematically described in Figure 2. A recent review by Griffith et al. lists the most common transporters and carriers overexpressed at the BBB and which can be potentially used by exogenous molecules to reach the brain parenchyma [43,44].

Due to their very diverse physicochemical nature, nanotherapies may be involved in a broad spectrum of transcytosis pathways (Figure 2). Furthermore, their tridimensional features almost consistently require the presence of a specific targeting entity on their surface to trigger an efficient brain translocation. A non-exhaustive list of these moieties is presented in Table 2, with some examples of nanomedicines that have been proven to effectively cross the BBB.

In addition to the widely used intravenous administration which requires a specific design of nanoparticles to target and bypass the BBB, another possible yet quite invasive route of administration is direct intracerebral drug infusion. This surgically assisted route of administration can either be achieved via stereotactic injection in the pathological area (i.e., the intraparenchymal route) in order to form local depots or via intraventricular administration [44]. Due to the extremely risky nature of these invasive administration methods, they are mainly restricted for life-threatening conditions and do not allow repetitive injections [45].

### 2.2. Intranasal Drug Delivery

Intranasal or nose-to-brain drug delivery is a very interesting route of administration for brain targeting, as it allows the formulation to bypass the BBB. This process can be achieved through two different nasal regions: the respiratory and the olfactory areas. The nasal epithelium harbors millions of olfactory neurons which enable the transport of agents directly to the brain through the olfactory bulb via transcellular diffusion [46,47]. The second pathway involves the respiratory epithelium which, through the trigeminal nerve, will also allow agents to directly reach the brain tissue [47]. The intranasal route has been examined with success in a variety of neuroinflammation-related diseases, as recently thoroughly reviewed by Rhea et al. [48].

### 2.3. Intracarotid Infusion

Intracarotid infusion technique consists of the administration of a drug or a fluid to the carotid artery, which is the main artery that carries blood from the heart to the brain [49]. Intracarotid drug delivery can instantaneously generate exceedingly high local concentrations of the novel pharmaceutical agent, thus assisting in regional delivery. This process has long been ignored for the development of drugs targeting CNS disorders, and the kinetics of such an administrative route is, to this day, poorly understood. However a few examples in the literature have shown that it can be a promising administration route, especially for the treatment of brain tumors [50,51].

### 2.4. Transmucosal Drug Delivery

Transmucosal drug delivery is an innovative and popular method of direct CNS drug delivery using heterotopic mucosal grafts which allows for the rapid uptake of a drug into the systemic circulation by avoiding first-pass metabolism [52,53,54]. This method relies on “mucoadhesion”, or adhesion to human mucosa, which presents extreme proximity to the blood circulation and can even allow the delivery of high-molecular-weight and polar agents to the CNS [55]. This technique can be adapted to products administered via nasal, oral, or vaginal routes with the use of sprays, gels, tablets, or even suppositories.

### 2.5. Physical or Pharmacological Disruption of the BBB

The physical integrity of the BBB can be partially and temporarily disrupted in order to enable the transient passage of therapeutic agents such as nanocarriers. These mechanical disruptions can be successfully performed by “physically” widening and opening tight junctions, with, for instance, the application of an external ultra-sound (US) source driving microbubbles across the BBB, or by using a magnetic gradient driving nano-ferrofluids across the BBB [56]. The use of pharmacological agents which can cause an osmotic imbalance (mannitol, fructose, milk amide, urea, or glycerol) or vasoactive (such as bradykinin, histamine, serotonin, glutamate) or inflammatory compounds, (namely, prostaglandins, some interleukins or tumor necrosis factor-α (TNF-α)) can also be an alternative to generate a momentary BBB opening [56,57].

Although very promising and efficient, these approaches are mostly based on the destruction of tight junctions and may involve permanent CNS damage, since they are not drug- or nanocarrier-specific. They might also allow the passage of a large variety of endo/exogenous (macro)molecules with the molecule or carrier of interest [56].

From left to right: (i) Paracellular diffusion—i.e., diffusion in between cells, limited by the presence of tight junctions formed between neighboring endothelial cells and used by a fraction of small hydrophilic molecules, as well as some viral and lipid-based particles; (ii) Transcellular diffusion—i.e., diffusion through the apical and basolateral endothelial membranes and across the intracellular space, mainly driven by favorable drug physicochemical properties, such as with molecules exhibiting hydrophobic surfaces or moieties (for instance, carmustine or temozolomide, and polymeric/lipid-based nanoparticles); (iii) Receptor-mediated transcytosis (RMT), which relies on the recognition between a ligand and an endothelial receptor at the luminal side of the BBB to mediate drug or nanocarrier translocation—this is the preferential way of entrance for many proteins, as well as most types of nanocarriers; (iv) Carrier-mediated diffusion (CMD), used to deliver small molecules (i.e., glucose, amino acids, and nucleotides) to the brain via protein carriers expressed at both the apical and basolateral endothelial membrane surfaces; (v) Adsorptive-mediated transport (AMT), based on the non-specific adsorption of cationic serum proteins to negatively charged domains of apical endothelial membranes, followed by translocation to the brain parenchyma—certain types of micelles and extracellular vesicles have been demonstrated to use this pathway.

## 3. CNS Disorders and Nanotherapeutics

### 3.1. Niemann–Pick Type C Disease—Nanocapturing Cholesterol

Niemann–Pick is a rare autosomal genetic disorder which affects approximatively 1:120,000 children globally and impacts, physiologically, the ability to metabolize and transport lipids and cholesterol within cells [66]. Consequently, an abnormal accumulation of fatty substances occurs within several tissues of the body including the brain, liver, spleen, bone marrow and, in more severe cases, the lungs. This usually translates into progressive damages and loss of functions, which progress into life-threatening complications possibly fatal [67]. Genetically, Niemann–Pick type C is caused by a mutation of the NPC1 or NPC2 gene (located on the long arm of chromosome 18 or 14, respectively), leading to NPC type 1C or NPC type 2C phenotypes, respectively. This pathology can occur at any age, whether it is during early childhood or in young adults (in mild cases, this pathology can go undiagnosed until early adulthood). However, it was uncovered to mainly affect children. As such, the symptoms will widely depend on the severity of the condition and might include, for instance, neurological and psychiatric disorders, cognitive loss, ataxia, cholestasis, hypotonia or hearing loss [67].

To this day, there is no cure available for Niemann–Pick type C disease, with the agent Miglustat, a synthetic, N-alkylated imino analogue of D-glucose, being the only drug approved as a first line of treatment of patients developing neurological disorders [68]. Nonetheless, several therapies are currently under massive investigation. One of the most promising is the use of a preparation of 2-hydroxypropyl-β-cyclodextrin (HPβCD), also known as VTS-270 (Mallinckrodt Pharmaceuticals) [69]. HPβCD is a FDA-approved excipient, with the capacity to capture cholesterol and therefore reduce its storage level in NPC1 cells in animal models. This formulation was found to significantly slow the development of neurological symptoms in patients with NPC1 disorder and is currently under clinical investigations in Phase IIb/III [70].

A major obstacle in the effective clinical translation of HPβCD is the huge amounts required in order to adequately clear the excess cholesterol. A recent study from Brown et al. showed that distearyl-phosphatidylethanolamine-polyethylene glycol-lipid micelles (DSPE-PEG-lipid micelles) of HPβCD could be a viable option in order to improve the delivery of this drug [71]. Indeed, using filipin-based and amplex red cholesterol assays, the authors showed that a 12 nm micelle formulation including DSPE-PEG significantly increased the HPβCD-mediated cholesterol efflux. In addition, the formulations containing the highest DSPE-PEG ratios were found to be the most potent, in spite of encapsulating a smaller amount of HPβCD. Even more surprisingly, treatment with only DSPE-PEG (without HPβCD) decreased cholesterol levels, showing potency on its own. As such, the authors postulated on a synergetic effect from both DSPE-PEG and HPβCD. To go further, they investigated autophagy flux by using TF.LC3 analysis and noticed an overall decrease number of autophagosomes and autolysosomes when exposed to the DSPE-PEG/HPβCD mixture in Npc1^−/−^ compared to Npc1^+/+^ cells. This suggested that a treatment with these agents facilitated cholesterol efflux while not impacting defects in autophagy.

### 3.2. Spinocerebellar Ataxia—Delivering VEGF-Mimicking Nanoconstructs

Ataxias are a subset of heterogenous neurological disorders which include both genetic and non-genetic types of the disease [72]. More specifically, Spinocerebellar ataxias (SCA) represent a large group of the pathological genetic form, with more than 40 different subtypes of SCA reported in the literature and a global prevalence of 3 in 100,000 [72,73,74]. This “spinocerebellar” neurodegenerative disease mainly affects the cerebellum, the peripheral nerves, the brainstem, the basal ganglia, and, in rare cases, the spinal cord. SCAs display a broad and complex genotype-phenotype spectrum, genetically caused by either repeat- or non-repeat mutations. Repeat expansion mutations often occur on the CGA nucleotide triplet, which encodes for polyglutamine. Consequently, pathogenic polyglutamine expansions translate directly into an abnormal extended polyglutamine, namely PolyQ, resulting in protein misfolding, PolyQ protein aggregation and ultimately cell toxicity/neurodegeneration [73]. Others SCAs caused by non-repeat mutations can be due to missense or nonsense mutations, insertions, or deletions.

Given the genetic heterogeneity of this pathology, many different symptoms and neuropathological alterations have been described with common clinical hallmarks to all SCAs being a loss of balance, altered coordination and slurred speech ability, while many SCAs also lead to premature deaths [72]. The manifestation of this disorder usually occurs in mid-adulthood but onset of SCA in early childhood or elderlies have also been reported, thus making SCA a disease which can cover the entire lifespan [75]. To this day, there is no cure available for SCAs, with medication focusing mostly on improving the quality of life for patients suffering from SCA. However, tremendous scientific progress has been made in both the understanding and diagnosis of this pathology.

As previously stated, SCA is a highly heterogeneous genetic disorder and the use of whole-exome and genome sequencing as a diagnostic tool showed great promises. Most notably, the use of nanotechnology can be of great interest in the case of SCA with for instance the Oxford nanopore sequencing system [76]. Engineered through a protein nanopore covalently attached to an adaptor molecule, this process enables identification of unlabeled nucleoside 5′ phosphate with an accuracy >99.8%, hence significantly facilitating the sequencing of large DNA sequences.

More recently, a study from Hu et al. showed that a synthetic, amphiphilic VEGF-mimicking peptide (synthetic VEGF mimetic peptide in which a 15-amino acid VEGF sequence (KLTWQELYQLKYKGI) is covalently linked to an amphiphilic peptide) able to self-assemble into nanoparticles (≈10 nm) could be a viable therapy to improve function in SCA1 [77,78]. Indeed, prior work from the same group proved that the levels of Vascular Endothelial Growth Factor (VEGF) are drastically diminished and practically abolished in SCA1 mice, which could be reversed by VEGF brain injection [79]. However, in their follow-up study, the authors admitted that the synthesis of recombinant VEGF is a very costly process, and that this protein is prone to proteolysis with a short half-life t_1/2_ < 30 min in vivo [78]. Therefore, the discovery and synthesis of nano-VEGF whose functional epitopes mimic the N-terminal domain of VEGF improved the SCA1 phenotype in several distinct aspects. This VEGF nanoformulation showed significant stability compared to recombinant VEGF, dramatically improved half-life (from 30 min to several days) and sustained release in vivo. These latter points are of particular interest when dealing with peptide therapy as they constitute major drawbacks to their direct clinical use [80,81]. In the case of Nano-VEGF, the in vivo release lasted more than 28 days, which is consistent with clinical applications. Even more importantly, Nano-VEGF significantly improved motor coordination, neuropathological measures and “firing” of Purkinje neurons (which functions are diminished in SCA1 mice) in mice with advanced SCA1 phenotypes. Of note, this agent was administered intracerebroventricularly (icv.) in order to avoid systemic effects, which could also explain the positive pharmacokinetic features of nano-VEGFs. Finally, the authors believe that this therapeutic strategy could be applied to other neurodegenerative diseases, such as amyotrophic lateral sclerosis (ALS), Parkinson’s and Alzheimer’s, which also display altered levels of VEGF. This approach could be further improved by designing VEGF-based nanocarriers displaying the capacity to effectively cross the BBB, as an iv. administration would be easier and safer than an icv. one.

### 3.3. Creatine Transporter Deficiency (CTD)—A Nasal Nanoemulsion for Brain Creatine Shipping

Cerebral creatine deficiency syndromes (CCDS) are a group of inborn errors of creatine metabolism which prevents the synthesis or transport of creatine in the human body [82]. Physiologically, creatine is required to insure normal levels of ATP both in the brain and other organs. Studies with CK knockout mice demonstrated clear cognitive loss, especially in the hippocampal region [83].

There are three distinct types of CCDS, including (i) creatine transporter deficiency (CDT), (ii) guanidinoacetate methyltransferase deficiency (GAMT), and (iii) arginine: glycine amidinotransferase deficiency (AGAT). The prevalence of these pathologies is difficult to assess, due to either the absence of studies (such as for AGAT) or conflicting reports; albeit, CCDS are known classified rare diseases [84]. Currently, it is estimated that CDT accounts for 1–2% of all unexplained X-linked intellectual disabilities, GAMT prevalence is estimated to be between 1 out of 2,640,00 and 1 out of 550,000 patients, while the prevalence of AGAT is unknown due to the lack of studies on records. CDT, on which we will mainly focus here, is an X-linked metabolic disorder caused by a defective creatine transporter [85]. This pathology, which was first described in a study of 2001, showed a complete absence of creatine signal in the brain but increased levels in both urine and plasma [85]. The altered creatine transporter, namely SLC6A9 (also known as CRTR or CT1), originates from a non-sense mutation which can affect both genders, with male carrying the hemizygous nonsense mutation while female will present the heterozygous form. Clinically, patients affected by CTD will suffer from severe language delay, cognitive impairments, intellectual disability, and behavioral disorders, such as attention deficit or hyperactivity, which can often be mistaken for autism. To date, there is no cure available for CTD, emphasizing the critical need for therapeutic alternatives. Usually, individuals who suffer from creatine deficiency are prescribed creatine supplementation as a first-line treatment (such as, for instance, in the case of AGAT). This helps improving quality of life and cognitive abilities. However, in the case of CTD, the BBB impedes the crossing of creatine (which is a positively charged polar small molecule whose guanidine moiety prevents from crossing the BBB) without the presence of a functional SLC6A9 transporter.

In 2019, a study from Ullio-Gamboa et al. addressed this challenge by synthesizing dodecyl creatine ester (DCE)-loaded nanoemulsion (with a size of 138–149 nm) [86]. This optimized formulation comprised 63% *w*/*w* Transcutol^®^ HP and 25% *w*/*w* of water, with an oil concentration which was maintained at 12% *w*/*w*, as well as DHA and Maisine ^®^ CC (2:1% *w*/*w*). This nanoemulsion was based on approved FDA excipients, offered several benefits including the protection of the DCE from enzymatic and chemical degradation. Interestingly, DCE nanoemulsions were also compatible with intranasal (in.) administration.

In Ullio-Gamboa et al.’s study, DCE is biotransformed in creatine by esterases through enzymatic route, a process which takes place in the brain tissue as esterases are normally present in endothelial cells [87]. The nasal administration of DCE nanoemulsions significantly enhanced cognitive impairments in vivo, improving novel object recognition (NOR) memory in *Slc6a8^−/y^* homozygous mice in comparison to the blank vehicle. Additionally, increased concentrations of creatine in different regions of the brain were described by the authors, while the ATP levels only increased in the mice striatum. These results suggest that DCE-loaded nanovesicles could be directly transported through the olfactory bulb/trigeminal nerve or indirectly via the lymphatic system, and subsequently to different brain regions, either extra-, intra- or transneuronaly. However, further detailed studies are required to understand the exact mechanism of transportation of DCE nanoemulsions in the brain. Of note, the authors announced that a follow-up study will include DCE-radiolabeled agents into the nanoengineered vesicles, which will help shed light on these unanswered questions.

### 3.4. Mucopolysaccharidosis Type I (MPS I)—A Nano Gene Therapy

Mucopolysaccharidosis type I (MPS I) is an autosomal recessive lysosomal disorder which originates from a deficiency in α-L-iduronidase (IDUA), an enzyme required for the proper degradation of two glycosaminoglycans (GAG)—namely, dermatan and heparan sulfate (HS) [88]. Therefore, insufficient IDUA activity leads to GAG and HS accumulation, ultimately resulting in severe neurological and musculoskeletal disorders [88].

Globally, the prevalence of this pathology is estimated to be 1 in 100,000 [89]. Scarpa et al. recently summarized a comprehensive overview of all treatment options for MPS pathologies, including hematopoietic stem cell transplantation, enzyme replacement therapy in the cerebrospinal fluid, and gene therapy. Despite the wide range of considered treatment options, the authors concluded that a significant, unmet medical need persists [90]. Historically, MPS I has been classified under 3 different syndromes: Hurler, Hurler-Scheie and Scheie [91]. Hurler syndrome, which accounts for 57% of cases, is known to be the most severe type of the pathology with significant cognitive impairments, joint stiffness, respiratory, heart and hepatic diseases, and most patients dying in early/late childhood. On the other hand, Scheie syndrome (which represent approximatively 20% of cases) usually affects patients through adulthood, show normal cognitive abilities but develop disease-related morbidity. Hurler-Scheie (≅23% of all cases) is an intermediate phenotype with patients experiencing mild or no cognitive impairments, while their life-expectancy is shortened due to the development of somatic symptoms in the second decade of their life [91]. Curative strategies are mostly palliative but current intervention include enzyme replacement therapies (ERT) and allogeneic hematopoietic stem cell transplantation (allo-HSCT), which have shown promise [92]. However, both also displayed limited efficacy in addition to several limitations, such as, for instance, the inability to cross the BBB and costly life-long infusions or transplant-associated mortality for ERT and allo-HSCT respectively. In this respect, non-viral gene therapy is an interesting alternative for patients suffering from MPS I, which can help overcome poor cell uptake and nucleic acid enzymatic degradation [93,94,95].

A first study of 2015 tackled this challenge with the synthesis of cationic nanoemulsions (NE) with a mean size ≈ 200 nm transfecting pIDUA (which is the plasmid carrying the gene encoding for IDUA) [96]. The authors, through their work, investigated several features, such as the physicochemical properties of the complexes and their transfection efficacy on a MPS I murine model. The cationic emulsions were composed of an Medium Chain Triglyceride (MCT) oil core stabilized by positively charged Dioleoylphosphatidylethanolamine (DOPE), Dioleoyl-3-trimethylammonium propane (DOTAP) and 1,2-distearoyl-*sn*-glycero-3-phosphoethanolamine-N-[amino(polyethylene glycol) (DSPE-PEG) complexing the negatively charged pIDUA. When the charge ratio was higher than +2/−, the synthetized nanoemulsions showed excellent stability and protection against nucleic acid enzymatic degradation. Additionally, the formulation with a charge ratio of +4/− displayed significant transfection efficacy in vitro in fibroblasts from MPS I patients. Furthermore, MPS I mice injected iv. with the NE-pIDUA complex showed an increase in IDUA activity in both the lungs and liver, which further confirmed the therapeutical interest of this method.

A follow up study from the same group investigated the formulation mechanism (adsorption or encapsulation of preformed pIDUA-DOTAP complexes into NE) with different charge ratios and proved a protective effect from deoxyribonuclease I degradation (DNAse I) [97]. In vivo transfection in the organs of MPS I mice injected iv. with NEP/pIDUA_A_ (associated with NE by adsorption) or NEP/pIDUA_E_ (associated with NE by encapsulation) showed increased IDUA activity in the lungs, liver, kidney and spleen; this activity was once again more potent for formulations with a higher charge ratio, thus confirming previous results (mean size ≈ 219–303 nm).

More recently, in 2018 the same group went further and explored the use of a liposomal formulation (with a mean diameter comprised between 81 and 108 nm) as nonviral carriers of the CRISPR/Cas9 system for both in vitro and in vivo gene therapy [98]. CRISPR/Cas9 technology, which relies on base pairing of nucleic acid, has revolutionized modern genome editing and its use combined with viral or non-viral delivery has already stimulated several research projects [99,100]. Overall, the authors showed that these complexes increased IDUA activity in several different organs, promoted long-lasting IDUA activity, normalized GAG levels, and effectively transfected mammalian cells. Therefore, gene editing using the CRISPR/Cas9 system could be of great interest to treat MPS I, while the authors also state that they will now be focusing on developing an approach which can be more relevant to reach the brain, primarily [98].

### 3.5. Rare Brain Infectious Diseases—How Nanomedicine Can Repurpose Clinically Approved Drugs

Free-living amoebas such as *Acanthamoeba*, *Balamuthia*, and *Naegleria* include species which have been described as potentially lethal to humans [101,102]. More specifically, amoebae pathogenic infections are classified as CNS-related rare diseases, causing two different and deadly types of encephalitis, namely primary amoebic encephalitis (PAM) or granulomatous amoebic encephalitis (GAE) [102,103,104]. GAE usually results from *Acanthamoeba spp.* and *Balamuthia mandrillaris* infections while *Naegleria fowleri* causes rapid meningeal inflammation, which is more commonly known as “brain-eating” PAM or Naegleriasis. In this last case, initial infection occurs when contaminated water is ingested by the host, which enables the invasion of the CNS via the nasal route [105,106]. Subsequent cerebral hemorrhage results in severe brain tissue damage within a matter of days only (5–7 days), and symptoms consist of impulsive onset of frontal or temporal headaches, fever, nausea, dizziness, hallucinations, and altered mental status, ultimately leading to coma and death in most cases. The mortality rate of Naegleriasis is estimated to be over 95%, highlighting the virulent nature of this pathology, especially in developing countries [107]. Until 2012, approximatively 310 cases have been reported globally, with 138 cases in the US between 1965 and 2015, of which 135 have been fatal [108]. Of note, a very recent study showed that even though the incidence of PAMs remains stable across the US, global warming and increased temperature might contribute to a northward expansion of *Naegleria fowleri* [109]. Moreover, the lack of awareness and diagnostic tools contribute even more to the significant mortality reported in the literature, while a misdiagnosis often worsen the chances of survival. The current therapeutic strategy consists of administering a cocktail of multiple drugs—e.g., amphotericin B (the most commonly drug used for PAM treatment), miltefosine, dexamethasone, ketoconazole, or sulfadiazine—which has shown success in a few cases reported in the literature [110,111]. However, when high concentrations of these drugs are administered intravenously (which is required due to the ability of the BBB to partially hamper these drugs from reaching the brain), severe side effects (such as, for instance, nephrotoxicity) can become a real threat to the patient’s life. Additionally, prolonged treatments with any type of antifungals or antibiotics can lead to resistance phenomena and drastically limit the potency of such agents. Therefore, there is a critical need to enhance the potency and brain penetration of antiamoebic drugs or to provide an alternative to the current treatments.

Between 2017 and 2018, two studies from Rajendran et al. and Anwar et al. explored the use of a metallic nanoparticle conjugation (mean size ≈ 20–100 nm) on the efficacy of three different drugs (amphotericin B, nystatin, and fluconazole, respectively) against *Naegleria fowleri* [112,113]. In order to do so, drugs conjugated with silver nanoparticle (AgNPs) were synthesized and their antiamoebic effects were evaluated versus drugs or NPs alone. The results revealed that conjugation with AgNPs significantly improved the potency of both amphotericin B and nystatin (whether *it was versus* the drug or NP alone) whereas fluconazole AgNPs showed only limited efficacy [112,113]. In addition, the authors studied host cell cytotoxicity by comparing host cell damage in presence of *Naegleria fowleri* or pretreated *Naegleria fowleri* which revealed very limited negative effects from AgNPs. Noteworthy, these results should be challenged and strengthened by a thorough evaluation of the in vivo effect of the drug alone versus conjugated with AgNPs, in a PAM phenotype, especially given the current concerns about AgNPs, related to the production of radicals/reactive oxygen species (ROS) [114]. Repurposing clinically FDA-approved drugs targeting CNS disorders can also be another effective drug discovery strategy. Therefore, another study published in *ACS Chemical Neuroscience* looked at the effect of diazepam, phenobarbital, and phenytoin alone or encapsulated in AgNPs against *N. fowleri* and other amoebae [115]. Benzodiazepines (such as diazepam) and barbiturates (such as phenobarbital) are known to target gamma-aminobutyric acid-A (GABA_A_) receptors, subsequently opening chloride ions channels and triggering hyperpolarization. which can lead to control over seizures [116,117]. On the other hand, phenytoin is also an anti-seizure drug which exerts its activity by blocking voltage-gated sodium ion channels [118]. Evaluation of these CNS-targeting drugs along with their silver nanoconjugates revealed promising amoebicidal potency against *N. fowleri*, with AgNPs conjugation significantly enhancing the overall activities [115].

More recently, a follow-up study from the same group looked at the effect of oleic acid-coated silver nanoparticles (OA-AgNPs) with a mean diameter comprised between 45 and 90 nm against *N. fowleri* [119]. OA is a naturally occuring monounsaturated omega-9 fatty acid, which has been associated with health benefits, such as hypotensive or anti-inflammatory effects; it also exhibited potent antiacanthamoebic activity against *Acanthamoeba castellanii* [120,121]. Thus, the use of OA against another amoebic strain such as *N. fowleri* made perfect sense and confirmed its broad antimicrobial spectrum of activity. Indeed, the authors noticed that OA-AgNPs significantly reduced amoebae activity compared to OA or AgNPs alone, while also drastically inhibiting *N. fowleri*-mediated host cell cytopathogenicity (65% decrease, approximatively). Furthermore, cytotoxicity assays revealed that OA, alone or conjugated with AgNPs, displayed a very safe profile, with a cytotoxic effect below 20%. Finally, the authors also postulated that OA might trigger apoptotic events in amoebae responsible for its antiamoebic effect.

With the goal of repurposing drugs effective against other acanthamoebic infections, future studies from the same group might include the use of metformin-coated silver nanoparticles against *N. fowleri* which were also found very recently to be potent against *Acanthamoeba castellanii* [122].

However, similarly to what was previously mentioned, in vivo testing in a relevant animal model is urgently required in order to give further consideration of Ag-NPs as viable alternatives to current PAM’s treatment.

### 3.6. Primary Central Nervous System Lymphoma (PCNSL)—Enhancing BBB Crossing

Primary central nervous system lymphoma is a highly aggressive, non-Hodgkin-type cancer which develops inside the CNS, including the brain, spine, cerebrospinal fluid (CSF) and the eyes [123]. This very rare type of tumor has an incidence of seven cases per 1,000,000 people in the US, with an overall survival of 12–16 months when diagnosed and treated (vs. 1.5-3 months when left untreated) [124]. Patients affected with PCNSL develop multiple neurologic disorders and behavioral changes, with symptoms such as focal neurologic deficits or headaches, nausea, papilledema, and seizures originating from an increased intracranial pressure. The diagnosis usually relies on brain MRI, CSF evaluation, stereotactic biopsy, and vitrectomy (in the presence of ocular involvement) in order to reveal the location, extent, and severity of the disease [123]. Chemotherapy with high-dose methotrexate (HD-MTX) is the first line of treatment and can be associated with radiation therapy [125]. However, it has been reported than more than 50% of patients suffering from PCNSL are subject to recurrence, and the use of systemic HD-MTX has long been described to cause severe adverse effects such as kidney failure [125].

There are currently 23 clinical trials investigating potential lines of treatments, with the use of other chemotherapeutic drugs being the main source of research in the field [126]. The contemporary focus relies on the optimization of current treatments and, in order to do so, clinicians looked closely at related tumors, such as, for instance, glioblastomas multiform (GBMs) or diffuse large B-cell lymphomas (DLBCLs). DLBCLs display many pathophysiological features resembling the ones of PCNSLs, and the standard treatment for this pathology is doxorubicin (DOX) [127]. On the other hand, and in spite of decades of massive investigations, GBMs remain one of the most deadly types of brain cancer, and the use of nanoencapsulated drugs has been suggested as a potential promising alternative [128,129].

As such, a study from 2019 tried to reconciliate these features by synthesizing conjugated nanoparticles loaded with DOX for the treatment of PCNSL [130]. In their study, the authors used an approach previously reported in the literature which aimed to target low-density lipoprotein receptor-related protein (LRP) [131]. These receptors were found to be overexpressed on the BBB and to mediate directly the transcytosis of several ligands across this barrier [132]. Angiopep-2, a specific ligand of LRPs, exhibiting strong brain penetration and transport capability, was used as a targeted group in this work, with the design of ANG-conjugated poly(ethylene glycol)-*b*-poly(ε-caprolactone) (PEG-*b*-PCL) nanoparticles (APP) loaded with the chemotherapeutic DOX agent (mean size ≈ 42 nm). Flow cytometry and confocal laser microscopy analysis showed that APP nanoparticles were efficiently internalized in the bEnd.3 brain cell line and that this process was thoroughly mediated by ANG. In addition, the authors performed MTT assays on the SU-DHL-2 lymphoma cell line, which showed the potent cytotoxic effect of APP DOX NPs but to a lesser extent compared with free DOX. Finally, and perhaps most interestingly, the in vivo effect of APP DOX-loaded nanoparticles was evaluated and showed significantly better survival times than PP DOX-loaded nanoparticles or free DOX, with a 26.1-day median survival vs. 19.2- and 18.1- for DOX-loaded nanoparticles and free DOX, respectively. This assay, performed on a PCNSL xenograft tumor model generated by intracranial injection of SU-DHL-2-Luc cells, also proved that the tumor area was significantly smaller after APP DOX treatment, suggesting that this formulation could be a viable therapeutic option for patients suffering from PCNSL.

Table 3 hereafter briefly summarizes the rare CNS disorders related to the development of nanotherapeutics within this work.

## 4. Future Directions and Conclusions

Thanks to novel better diagnostic tools and improved disease knowledge, a multitude of rare CNS disorders can now be clearly characterized. As a consequence, this last decade has seen a surge in the discovery of new and rare pathologies which have never before been reported in the literature. This process, however, goes along with challenges for the scientific community, which is left without relevant, conventional therapeutic tools. Therefore, this area provides a whole new direction to research and development in this field, with many notable advances achieved during these past few years. Targeted gene therapy, such as CRISPR/Cas9, holds great promises and has emerged as a powerful technology, especially for rare diseases. However, many therapeutic targets cannot currently be efficiently accessed. These limitations, in addition to the “hindering” BBB, prompt for the development of non-viral carriers which can assume the transport, protection, and delivery of a wide range of molecular or sub-molecular complexes. Therefore, the use of nanotechnology, and more specifically nanocarriers, shows great versatility, and their ability to cross or bypass the BBB is still of tremendous importance.

Nonetheless, a noteworthy setback impeding the development of nanomedicinal approaches is the almost complete lack of information on the cellular and tissular interactions of such nanocarriers, other than those taking place with the targeted tissue. Despite enthusiastic reports on the efficiency of these nanovectors in crossing the BBB and reaching the brain parenchyma in significant concentrations, to date very few studies include data on the immunogenicity, inflammation potential, or genotoxicity of these nanoobjects. These missing data not only affect nanocarriers designed to treat rare CNS pathologies but significantly impact the potential clinical translation of all these brain-related nanotherapies. This is why a marked and sustained effort from the scientific community must be undertaken to overcome this “gap in knowledge” and ensure a viable future for these nanotherapeutic alternatives.

Finally, the access to such technologies can also be a high-cost burden, despite renewed efforts from worldwide health entities and patient associations to financially support drug discovery and novel therapeutic strategies. This is especially true in developing countries, where a non-negligeable number of orphan/rare diseases are reported. This highlights even more the relevance of nanotherapeutics as diagnostic tools, with several CNS rare diseases showing a much better outcome when identified and treated early.

As such, we hope that our review will help in stimulating efforts from the scientific community to further identify specific probes, biomarkers, or nanocarriers which could help in expanding our knowledge on rare neurological diseases and nanomedicine.

## Figures and Tables

**Figure 1 pharmaceuticals-14-00109-f001:**
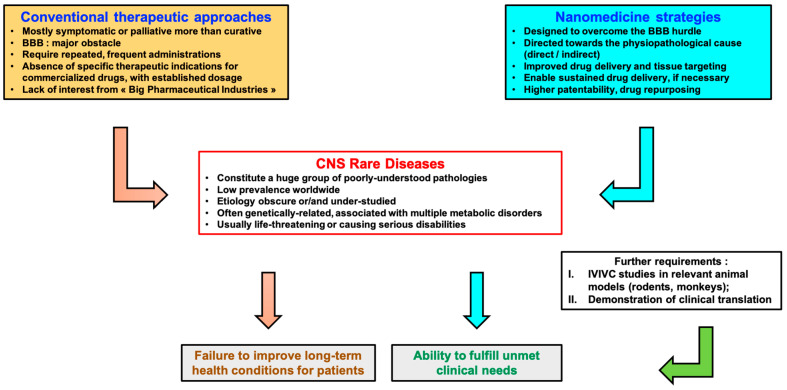
Comparative scheme representing the curative benefits of nanomedicine strategies vs. conventional therapeutics.

**Figure 2 pharmaceuticals-14-00109-f002:**
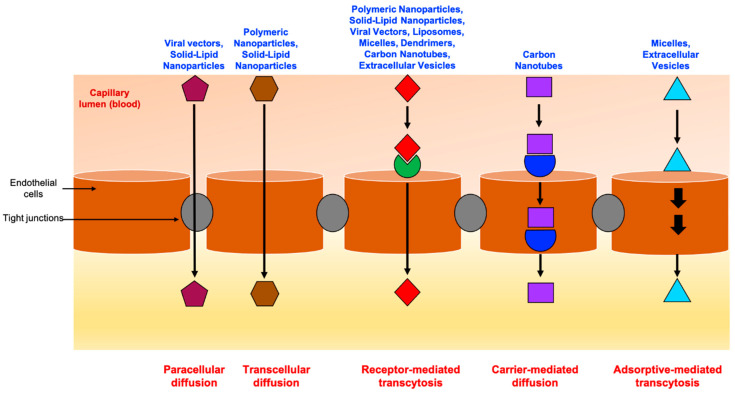
Transport routes across the BBB.

**Table 1 pharmaceuticals-14-00109-t001:** Nanocarriers currently developed for the treatment of CNS-related disorders.

Origin	Nanocarrier Designation	Size Range * (nm)	Primary Material	Benefits (+) and Disadvantages (−)	Ref.
Biological	Viral vectors	<100	Viral capsid proteins	(+) High transfection efficiency (−) High immunogenicity, high production costs, important adverse effect risks	[31]
	Extracellular vesicles	50–500	Plasma membrane phospholipids	(+) Good safety profile, enhanced specific targeting, controlled pharmacokinetics (−) Very limited data, inadequate in vivo targeting efficiency	[32]
Biologically mimicking	Liposomes	20–250	Phospholipids	(+) Drug protection, passive diffusion across the BBB, adapted for both hydrophilic and hydrophobic drugs, possible specific targeting (−) Potential (neuro)toxicity, physicochemical instability, clearance issues	[33]
Chemical	Micelles	10–100	Surfactants	(+) Absence of neurotoxicity, enhanced drug bioavailability, physicochemical stability, ability to control drug release (−) For lipophilic drugs only, poor drug loading efficiency, physicochemical instability	[34]
	Lipid nanoparticles (LNPs)	<100	Cationic lipids	(+) Improved drug loading of genetic material, stability (−) Immunogenicity issues, do not passively target BBB, rapidly cleared from blood	[35,36]
	Solid lipid nanoparticles (SLNs)	50–500	Solid lipids	(+) Biocompatibility and degradability, absence of neurotoxicity, drug protection, improved control drug release, ability to cross the BBB via passive diffusion (−) Reduced loading efficiency for hydrophilic drugs	[37]
	Dendrimers	<10	Organic dendrons	(+) Adapted for both hydrophilic and hydrophobic drugs, enhanced specific targeting, physicochemical stability (−) Potential (neuro)toxicity, clearance issues, potential organ accumulation	[38]
	Polymeric nanoparticles	<500	Synthetic or natural polymers	(+) Biocompatibility, possible biodegradability, drug protection, ability to control/sustained drug release, enhanced specific targeting (−) Potential (neuro)toxicity (neuroinflammation, neurodegeneration)	[39]
	Carbon nanoparticles (nanotubes CNT, quantum dots QD)	<10 (QD) <100 (CNT)	Carbon	(+) Specific chemical, mechanical, and electrical properties, accumulation in brain tissue, enhanced surface functionalization (−) (Neuro)toxicity issues (absence of degradation, accumulation)	[40,41]
	Inorganic nanoparticles	2–100	Au, Ag, ZnO, Si, ceramic NPs, superparamagnetic iron oxide NPs (SPIONs)	(+) Electrical, mechanical and optical properties, high surface area useful for grafting targeting moieties (−) Established (neuro)toxicity, requires prior functionalization to cross the BBB	[42]

* most usual hydrodynamic diameters, as observed by Dynamic Light Scattering (DLS).

**Table 2 pharmaceuticals-14-00109-t002:** Most common targeting moieties allowing agents to bypass the BBB.

Targeting Moiety	Endothelial Target	Examples of ‘Nano’ Applications	Ref.
Transferrin (Tf), lactoferrin (Lf), anti-TfR antibodies or aptamers	Tf receptors (TfR)	Lipid SPION nanovectors loaded functionalized with antibodies against the transferrin receptor	[58]
		TfR-binding Fc polypeptide	[59]
Insulin	Insulin receptor	Insulin PEG-coated gold particles	[60]
ApoB, ApoE, Angiopep-2	Low density lipoprotein receptors (LDLR)	ApoE2 loaded brain-targeted functionalized-liposomes	[61]
		Angiopep-2 Peptide-Modified Bubble Liposomes	[62]
		PLGA-PEG-Ang–2 nanoparticles	[63]
FC5	Cell surface α(2,3)-sialoglycoprotein (namely TMEM-30A)	FC5 bivalently fused with human Fc domain	[64]
Arginine-Glycine-Aspartic (RGD) peptide	Integrin receptors	RGD peptide-modified ultrasmall Au-ICG nanoparticles	[65]

**Table 3 pharmaceuticals-14-00109-t003:** Nanotherapeutic approaches currently reported for rare CNS disorders.

Strategy		Summary	Example(s) of Treated Pathology	Biological Model or Clinical Stage	References
Suppress Cause of Disease	Gene therapy	Nanocarriers can deliver genes in a safer, non-viral way. Genes will then express the absent or deficient protein in therapeutic levels. They can also “complex” or “load” excess of substances causing brain damages due to proper physicochemical engineering.	Niemann–Pick type C disease	Phase IIb/III	[70]
	Capture therapy		Mucopolysacchari-dosis type I (MPS I)	In vivo (mice); ex vivo (MPS I patients fibroblasts)	[96,97,98]
Target Cell Defect		Nanoencapsulation may allow the delivery of intact peptides or proteins by protecting them against chemical and biological degradations.	Spinocerebellar ataxia	In vivo (mice) and ex vivo (ischemic limb model)	[77,78]
Supplement Deficiency Protein/Peptide Delivery		Nanoconstructs can be engineered to mimic deficient proteins or peptides in order to restore a defective cellular pathway.	Creatine transporter deficiency (CTD)	In vivo (mice)	[86]
Repurpose Clinically Approved Drugs		Many currently approved drugs could be efficient against CNS rare diseases. Nanotechnology enables repurposing their use in such pathological conditions, while improving efficiency and reducing systemic side effects.	Rare brain infectious diseases	In vitro	[112,113,115,119,122]
Enhance BBB Passage		Nanocarriers can be decorated with ligands specifically binding with receptors at the surface of brain endothelial cells. Following ligand–receptor binding, transcytosis is mediated and drugs are delivered to the brain tissue in an enhanced fashion.	Primary central nervous system lymphoma (PCNSL)	In vivo (mice)	[130]

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
