# Peer review of "Think Big, Start Small: How Nanomedicine Could Alleviate the Burden of Rare CNS Diseases"

_pharmaceuticals, 2021, doi:10.3390/ph14020109_

Round 1

Reviewer 1 Report

The review manuscript "Think big, start small: how nanomedicine could alleviate the burden of rare CNS diseases" presents an overview of some of the rare CNS diseases, discussing their physiopathology, and describing, briefly, the limitations and benefits that nanotechnologies could offer to treat these pathologies. Finally, it address some issues that can hinder the development of efficient therapies in this area.

The review manuscript is organized in three different sections: Introduction, CNS disorders and nanotherapeutics, and Future directions and conclusions; it is well written and the English used is correct and readable.

The manuscript could be suitable for publication on “Pharmaceuticals”, but some major and minor revisions have to be considered before publication.

Major revisions:

- Taking into account that: a) the manuscript was thought and written to be published in “Pharmaceuticals”, and b) one of the aims of the authors is focusing it on how nanomedicine could alleviate the burden of rare CNS diseases, a brief section, in which the authors mention and describe concisely the current different vectors/nanocarriers that exist for this purpose, should be included. With the advantages and disadvantages of each type.

Moreover, when the authors describe the different reports (Section 2), a more detailed, but not exhaustive, (chemical) description about the nanosystem used in each report, should be presented.

- Since one of the focus of the manuscript is to describe nanosystems/nanotherapies/nanomedicines that can pass the BBB to treat rare CNS disorders, a section with a short description on the CNS barriers and possibilities to overcome them should be introduced in the current manuscript. As well as some moieties that can be explored to target the described CNS disorders, and administration routes. All of them are major concerns when one designs nanomedicines.

Minor revisions:

- The authors should rethink about the way of exposing/organizing the information in the abstract. They present firstly the most commonly neurodegenerative disorders (Alzheimer’s and Parkinson’s diseases) and just as “in the other hand”, the unrenowned CNS disorders are introduced. Being that the topic of this manuscript are these rare diseases, this way of presentation is a little bit confusing.

- Lines 58-59: The authors say “First of all, these agents can be designed in such a way that allows them to efficiently cross the BBB”.

Can the authors elaborate, briefly, about the properly design of nanocarriers/nanosystems to overcome the mentioned obstacles to cross the BBB?

- In the introduction (lines 52-54), the authors mention that the presence of very tight, selective junctions created by the BBB allows the crossing of small polar and hydrophobic agents. However, in the lines 254-256, they state that “in the case of CTD, the BBB impedes the crossing of creatine (which is a positively-charged small molecule) without the presence of a functional SLC6A9 transporter”. Since the creatine is a “small polar molecule”, this seems to contradict the previous sentence (lines 52-54). Could you the authors clarify this issue?

- Lines 331-332: The size of the liposomal formulations is given (“between 81 and 108 nm”), however this information is missed for the complexes previously mentioned in the same section 2.4. Why do the authors consider important to stress the size in this case, but not in the previous ones?

Taking into account the necessity of BBB crossing of the nanocarriers/nanosystems for the treatment of any disorder related to the CNS, their size is very important. Thus, it would be relevant to give this information for all described cases.

- The authors give the incidence for some of the reported rare diseases, however in other cases (e.g. cerebral creatine deficiency syndromes and rare brain infectious diseases) this information is missed. The authors should try to maintain the coherence of the given information in all cases.

- The same for the journal in which the described reports were published. Sometimes the name of the journal is given, some other times is not.

- The authors could update the reference 2 (from 2012) using more recent reference. Some examples:

a) Spencer AP, et al. Breaking Barriers: Bioinspired Strategies for Targeted Neuronal Delivery to the Central Nervous System. Pharmaceutics. 2020 Feb 23;12(2):192. doi: 10.3390/pharmaceutics12020192. PMID: 32102252; PMCID: PMC7076453.

b) Soni S, et al. Nanomedicine in Central Nervous System (CNS) Disorders: A Present and Future Prospective. Adv Pharm Bull. 2016;6(3):319-335. doi:10.15171/apb.2016.044.

c) Mukherjee S, et al. Recent Advancements of Nanomedicine in Neurodegenerative Disorders Theranostics. Adv Funct Mat 2020. Doi: 10.1002/adfm.202003054.

- Lines 313-314: “The cationic emulsions were composed of an MCT (Medium Chain Triglycerides) oil core stabilized by…”. It should appear first the full name and then the abbreviation in brackets (as done in the manuscript in general).

Author Response

We would like to thank the reviewer for his/her constructive criticisms. All changes made in the text have been highlighted in yellow.

A point-by-point rebuttal of concerns raised by this reviewer can be found below.

Major concerns :

1- Taking into account that: a) the manuscript was thought and written to be published in “Pharmaceuticals”, and b) one of the aims of the authors is focusing it on how nanomedicine could alleviate the burden of rare CNS diseases, a brief section, in which the authors mention and describe concisely the current different vectors/nanocarriers that exist for this purpose, should be included. With the advantages and disadvantages of each type.

In accordance with the reviewer's recommendation, we added a table (now Table 1) summarizing the various types of nanocarriers presently developed and used to treat CNS diseases. Advantages and disadvantages of each type are also indicated in this table.

2- Moreover, when the authors describe the different reports (Section 2), a more detailed, but not exhaustive, (chemical) description about the nanosystem used in each report, should be presented.

Details about the physicochemical features of each nanosystem presented in ex-section 2 (curently section 3) were added throughout the text and highlighted in yellow for the reviewer's convenience.

3- Since one of the focus of the manuscript is to describe nanosystems/nanotherapies/nanomedicines that can pass the BBB to treat rare CNS disorders, a section with a short description on the CNS barriers and possibilities to overcome them should be introduced in the current manuscript. As well as some moieties that can be explored to target the described CNS disorders, and administration routes. All of them are major concerns when one designs nanomedicines.

We amended this part of the manuscript by creating Section 2 : Nanocarriers used for the treatment of CNS disorders and strategies to bypass the BBB (lines 146 to 242) and adding Figure 1, which describes the different types of nanocarriers and the way they cross  the BBB. Table 2 was also added, with examples of the commonest targeting moeties used to translocate drugs and nanocarriers across the BBB to the brain parenchyma.

Minor concerns :

4- The authors should rethink about the way of exposing/organizing the information in the abstract. They present firstly the most commonly neurodegenerative disorders (Alzheimer’s and Parkinson’s diseases) and just as “in the other hand”, the unrenowned CNS disorders are introduced. Being that the topic of this manuscript are these rare diseases, this way of presentation is a little bit confusing.

We agree with the reviewer that this part of the abstract was poorly phrased. The abstract was modified accordingly with a more pronounced focus on CNS rare diseases and nanotechnologies (lines 19–21).

5- Lines 58-59: The authors say “First of all, these agents can be designed in such a way that allows them to efficiently cross the BBB”. Can the authors elaborate, briefly, about the properly design of nanocarriers/nanosystems to overcome the mentioned obstacles to cross the BBB?

We amended this part of our review with a brief statement and examples (lines 53-55) and the addition of Table 2, as well as a specific paragraph (lines 156 -171).

6- In the introduction (lines 52-54), the authors mention that the presence of very tight, selective junctions created by the BBB allows the crossing of small polar and hydrophobic agents. However, in the lines 254-256, they state that “in the case of CTD, the BBB impedes the crossing of creatine (which is a positively-charged small molecule) without the presence of a functional SLC6A9 transporter”. Since the creatine is a “small polar molecule”, this seems to contradict the previous sentence (lines 52-54). Could you the authors clarify this issue ?

The reviewer raises a very interesting point. Even though creatine is a small polar molecule, it is very positively charged due to the presence of a guanidine moiety on its chemical structure and has the ability to make numerous hydrogen bonds which hinders how it crosses the BBB. This was added to the description of creatine (lines 364-366).

7- Lines 331-332: The size of the liposomal formulations is given (“between 81 and 108 nm”), however this information is missed for the complexes previously mentioned in the same section 2.4. Why do the authors consider important to stress the size in this case, but not in the previous ones? Taking into account the necessity of BBB crossing of the nanocarriers/nanosystems for the treatment of any disorder related to the CNS, their size is very important. Thus, it would be relevant to give this information for all described cases.

We amended this part of our review by providing mean sizes or diameters for each type of formulation described in this manuscript (modifications highlighted in yellow in the corresponding sections).

8- The authors give the incidence for some of the reported rare diseases, however in other cases (e.g. cerebral creatine deficiency syndromes and rare brain infectious diseases) this information is missed. The authors should try to maintain the coherence of the given information in all cases.

We amended this part of our review by providing the incidence of CDT and rare brain infectious diseases to the manuscript (added information highlighted in yellow in the corresponding sections).

9- The same for the journal in which the described reports were published. Sometimes the name of the journal is given, some other times is not.

The names of Journals in which the authors published their work were deleted.

10- The authors could update the reference 2 (from 2012) using more recent reference.

Reference 2 was updated accordingly with Spencer, A.P.; Torrado, M.; Custódio, B.; Silva-Reis, S.C.; Santos, S.D.; Leiro, V.; Pêgo, A.P. Breaking Barriers: Bioinspired Strategies for Targeted Neuronal Delivery to the Central Nervous System. Pharmaceutics 2020, 12, 192, doi:10.3390/pharmaceutics12020192.

11- Lines 313-314: “The cationic emulsions were composed of an MCT (Medium Chain Triglycerides) oil core stabilized by…”. It should appear first the full name and then the abbreviation in brackets (as done in the manuscript in general).

This typo was amended accordingly so that we maintain homogeneity throughout the entire text (line 418).

Reviewer 2 Report

In my opinion the manuscript is well written, the main subject is well and clearly described. However, there are some issues which should be address before acceptance.

1
Very often in many publication we can find very enthusiastic information concerning the possibilities of modern nanomedicine, nano pharmacology and drug delivery. But the information is not fulfilling the entire problem. Very often the presented outcomes are limited to the description of the interaction of synthesized nanoparticles in the place of the destination. Unfortunately, there are a lot of missing data, for example no data concerning interaction with very important cells like even cells of immunological system etc, no information about possibilities of new nano carriers to turn on / promote the inflammation process etc., no information about genotoxicity, about in vivo interactions etc. I suggest adding the paragraph in which the issue will be discussed. The main idea of the paragraph should focused on the showing how much efforts need to be overtaken to obtain real well defined nanocarrier dedicated for drug delivery.

2
Currently, development of nano strategy for targeted drug delivery to brain tissue is very popular. Of course, it is very promising and important issue. But we should be aware that the delivery should be realized without affecting the structure and function of the BBB. It is very important issue and need to be discussed in the manuscript because very often we can find the publications in which authors show the data concerning novel nano formulations which can cross BBB but …. Unfortunately they use some chemicals which cover particles and cause unsealing of BBB, which is, in my opinion, unacceptable.

Author Response

We would like to thank the reviewer for his/her constructive criticisms. All changes made in the text have been highlighted in yellow.

A point-by-point rebuttal of concerns raised by this reviewer can be found below.

1- Very often in many publication we can find very enthusiastic information concerning the possibilities of modern nanomedicine, nano pharmacology and drug delivery. But the information is not fulfilling the entire problem. Very often the presented outcomes are limited to the description of the interaction of synthesized nanoparticles in the place of the destination. Unfortunately, there are a lot of missing data, for example no data concerning interaction with very important cells like even cells of immunological system etc, no information about possibilities of new nano carriers to turn on / promote the inflammation process etc., no information about genotoxicity, about in vivo interactions etc. I suggest adding the paragraph in which the issue will be discussed. The main idea of the paragraph should focused on the showing how much efforts need to be overtaken to obtain real well defined nanocarrier dedicated for drug delivery.

This is a very important point raised by the reviewer and we completely agree with this position. We therefore added a sub-section concerning this problematic in the conclusion & perspectives (lines 579-588).

2- Currently, development of nano strategy for targeted drug delivery to brain tissue is very popular. Of course, it is very promising and important issue. But we should be aware that the delivery should be realized without affecting the structure and function of the BBB. It is very important issue and need to be discussed in the manuscript because very often we can find the publications in which authors show the data concerning novel nano formulations which can cross BBB but …. Unfortunately they use some chemicals which cover particles and cause unsealing of BBB, which is, in my opinion, unacceptable.

The reviewer raises once more a very important point. This issue was amended and discussed in the introduction (lines 85-95).

Reviewer 3 Report

The authors provide an interesting review on the necessity of the discovery of new therapies for rare CNS diseases and how nanotechnology offers a feasible alternative for it. The review includes specific rare diseases, recent literature addressing their treatments through nanosystems. I found the review really helpful and stimulating to readers to focus on this problematic. I really congratulate the authors for this review.

I recommend just a minor revision of the review, only some details are missed and I would like the authors to include some of their opinions concerning specific topics.

-Authors point out that intravenous is an invasive drug delivery. In my opinion, direct intraventricular and intracerebral administrations are invasive methodologies but intravenous is not such invasive. It can be discomforting but not invasive. Sure other methodologies such intranasal or dermal deliveries are non-invasive but intravenous, talking about CNS drug delivery I would not consider it an invasive technique.

- Recent reviews also emphasize the potential of different nanotechnologies for CNS diseases (more common or rare diseases). It would be helpful for the readers to include them in this review. Eg. Samaridou et al, Bioorg Med Chem. 2018 Jun 1;26(10):2888-2905. doi: 10.1016/j.bmc.2017.11.001, Rodriguez-Otormin et al, Wiley Interdiscip Rev Nanomed Nanobiotechnol. 2019 Jan;11(1):e1532. doi: 10.1002/wnan.1532. Epub 2018 Jun 14, Dev Jayant et al, Expert Opin Drug Deliv. 2016 Oct; 13(10): 1433–1445.

- An extra problematic for treatments in these diseases, apart from the funding which is clearly the major issue, is the lack of realistic models where to test the nanodevices (in general for all the BBB affected diseases. Do the authors have an opinion on this issue?

Author Response

We would like to thank the reviewer for his/her encouragements, as well as constructive criticisms. All changes made in the text have been highlighted in yellow.

A point-by-point rebuttal of concerns raised by this reviewer can be found below.

  1. Authors point out that intravenous is an invasive drug delivery. In my opinion, direct intraventricular and intracerebral administrations are invasive methodologies but intravenous is not such invasive. It can be discomforting but not invasive. Sure other methodologies such intranasal or dermal deliveries are non-invasive but intravenous, talking about CNS drug delivery I would not consider it an invasive technique.
  • We amended our statement in the manuscript (line 95 and line 158).
  1. Recent reviews also emphasize the potential of different nanotechnologies for CNS diseases (more common or rare diseases). It would be helpful for the readers to include them in this review. Eg.Samaridou et al, Bioorg Med Chem. 2018 Jun 1;26(10):2888-2905. doi: 10.1016/j.bmc.2017.11.001, Rodriguez-Otormin et al, Wiley Interdiscip Rev Nanomed Nanobiotechnol. 2019 Jan;11(1):e1532. doi: 10.1002/wnan.1532. Epub 2018 Jun 14, Dev Jayant et al, Expert Opin Drug Deliv. 2016 Oct; 13(10): 1433–1445.
  • The manuscript was amended with addition of all the references here above (now references 97, 5 and 86, respectively).
  1. An extra problematic for treatments in these diseases, apart from the funding which is clearly the major issue, is the lack of realistic models where to test the nanodevices (in general for all the BBB affected diseases. Do the authors have an opinion on this issue?
  • This issue was amended and discussed in the introduction (lines 85-95).

Round 2

Reviewer 1 Report

This version of the manuscript has been significantly improved. The authors have revised and answered to the majority of the reviewer's comments.

However, I still have some comments before publication:

- Lines 58-59: The authors say “First of all, these agents can be designed in such a way that allows them to efficiently cross the BBB”. Can the authors elaborate, briefly,about the properly design of nanocarriers/nanosystems toovercome the mentioned obstacles to cross the BBB?

We amended this part of our review with a brief statement and examples (lines 53-55) and the addition of Table 2, as well as a specific paragraph (lines 156 -171).

Besides the answer referred by the authors and Table 2, they should mention that the suitability of nanotherapeutics to cross the BBB, derives from the numerous possibilities of tuning their characteristics (size, shape, surface properties, hydrophobicity, drug encapsulation or conjugation capacity and releasing – the latter already mentioned by the authors), as well as the possibility of functionalizing the nanocarriers with targeting moieties (this already mention by the authors) to improve their efficacy in reaching the desired tissue and/or cells.

- Table 1. The table needs a significant improvement. Not only regarding organization (for instance, viral vectors should be placed firstly and, then, to organize the non viral vectors following a reasoned/logic order), but also regarding contents and information. For example, one issue is the type of particles (nanoparticles, micelles…) and another one is the type of materials/(macro)molecules (polymers, dendrimers, lipids…) in which these particles are based on. Both of them can be lipid-, polymer- o dendrimer-based.

Liposomes and solid-lipid nanoparticles are bot lipid-based nanoparticles. And there are another lipid-based nanoparticles that could be included in this table. The same with the inorganic nanoparticles.

The benefits and disadvantages should be also revised. For instance, no all used polymers are biodegradable, they have even better possibility than liposomes for specific targeting, etc.

Moreover, there are some conceptual errors. E.g. “carbon nanotubes”, as the name indicates, are organic, no inorganic.

- Table 2. In this table, SPION nanoparticles are mentioned for first time. So, the full name is required. And also, these type of nanoparticles should be included in Table 1, within the corresponding category (inorganic nanoparticles).

Author Response

We thank the reviewer for acknowledging the improvements made according his/her criticisms and for pointing out the remaining weaknesses of this manuscript. We have now complied to all of the reviewer's remarks.

1- Besides the answer referred by the authors and Table 2, they should mention that the suitability of nanotherapeutics to cross the BBB, derives from the numerous possibilities of tuning their characteristics (size, shape, surface properties, hydrophobicity, drug encapsulation or conjugation capacity and releasing – the latter already mentioned by the authors), as well as the possibility of functionalizing the nanocarriers with targeting moieties (this already mention by the authors) to improve their efficacy in reaching the desired tissue and/or cells.

The passage mentioned by the reviewer has been rephrased and improved according recommendations (lines 53-61, highlighted in green).

2- Table 1. The table needs a significant improvement. Not only regarding organization (for instance, viral vectors should be placed firstly and, then, to organize the non viral vectors following a reasoned/logic order), but also regarding contents and information. For example, one issue is the type of particles (nanoparticles, micelles…) and another one is the type of materials/(macro)molecules (polymers, dendrimers, lipids…) in which these particles are based on. Both of them can be lipid-, polymer- o dendrimer-based.

Liposomes and solid-lipid nanoparticles are bot lipid-based nanoparticles. And there are another lipid-based nanoparticles that could be included in this table. The same with the inorganic nanoparticles.

The benefits and disadvantages should be also revised. For instance, no all used polymers are biodegradable, they have even better possibility than liposomes for specific targeting, etc.

Moreover, there are some conceptual errors. E.g. “carbon nanotubes”, as the name indicates, are organic, no inorganic.

Table 1 has been completely 'redesigned', with a presentation based on classification, as suggested by the reviewer. Columns have been added (origin, size range, primary material) so as to provide more information. Nanocarrier types have been added (Lipid nanoparticles, Carbon nanoparticles) and errors pointed out by the reviewer corrected. Table 1 is highlighted in green for greater convenience.

3- Table 2. In this table, SPION nanoparticles are mentioned for first time. So, the full name is required. And also, these type of nanoparticles should be included in Table 1, within the corresponding category (inorganic nanoparticles).

As suggested by the reviewer, SPION were added to Table 1, in the inorganic nanoparticles section and their full name disclosed there. Therefore the term SPION was left unchanged in Table 2, as explications were given in Table 1.

Reviewer 2 Report

I recommend the manuscrip for publication.

Author Response

We thank the reviewer for the accomplished revision work.